# Evaluation of the diagnostic performance of EpiTuub® Fecal Rotavirus Antigen Rapid Test Kit in Amhara National Regional State, Ethiopia: A multi-center cross-sectional study

Debasu Damtie[1,2]*, Aschalew Gelaw[3], Yitayih Wondimeneh[3], Yetemwork Aleka[1], Zewdu Siyoum Tarekegn[4], Ulrich Sack[5], Anastasia N. Vlasova[6,7], Belay Tessema[3,5]

1 Department of Immunology and Molecular Biology, School of Biomedical and Laboratory Sciences, College of Medicine and Health Sciences, University of Gondar, Gondar, Ethiopia, 2 Ohio State University Global One Health Initiative LLC, Eastern Africa Regional Office, Addis Ababa, Ethiopia, 3 Department of Medical Microbiology, School of Biomedical and Laboratory Sciences, College of Medicine and Health Sciences, University of Gondar, Gondar, Ethiopia, 4 Department of Veterinary Paraclinical Studies, College of Veterinary Medicine and Animal Sciences, University of Gondar, Gondar, Ethiopia, 5 Institute of Clinical Immunology, Faculty of Medicine, University of Leipzig, Leipzig, Germany, 6 Center for Food Animal Health, Department of Animal Sciences, College of Food Agricultural and Environmental Sciences, The Ohio State University, Wooster, OH, United States of America, 7 Department of Veterinary Preventive Medicine, The College of Veterinary Medicine, The Ohio State University, Wooster, OH, United States of America

☯ These authors contributed equally to this work.
* debidam@gmail.com

## Abstract

Rotavirus is the leading cause of morbidity and mortality due to acute gastroenteritis among children under five years globally. Early diagnosis of rotavirus infection minimizes its spread and helps to determine the appropriate management of diarrhea. The aim of this study was to evaluate the performance of EpiTuub® Fecal Rotavirus Antigen Rapid Test Kit for the diagnosis of rotavirus infection among diarrheic children under five years in Ethiopian healthcare settings. A total of 537 children with diarrhea were enrolled from three referral hospitals in Amhara National Regional State, Ethiopia. The samples were tested using one-step RT-PCR and EpiTuub® Fecal Rotavirus Antigen Rapid Test Kit (KTR-917, Epitope Diagnostics, San Diego USA) in parallel. Diagnostic performance of the rapid test kit was evaluated using the one-step RT-PCR as a gold standard. The sensitivity, specificity, and predictive values of the rapid test kit were determined. Moreover, the agreement of the rapid test kit with one step RT-PCR was determined by kappa statistics and receiver operators' curve (ROC) analysis was done to assess the overall diagnostic accuracy of the rapid test kit. Fecal Rotavirus Antigen Rapid Test Kit has shown a sensitivity of 75.5% and specificity of 98.2%. The kit was also found to have 89.9% and 95.0% positive and negative predictive values, respectively. The Fecal Rotavirus Antigen Rapid Test Kit has shown a substantial agreement (78.7%, p = 0.0001) with one-step RT-PCR. The overall accuracy of the Fecal Rotavirus Antigen Rapid Test Kit was excellent with the area under the ROC curve of 86.9% (95% CI = 81.6, 92.1%) (p = .0001). Thus, Fecal Rotavirus Antigen Rapid Test is a sensitive, specific, user-friendly, rapid, and equipment-free option to be used at points of care in Ethiopian health care settings where resource is limited precluding the use of one step RT-PCR.

**Data Availability Statement:** All relevant data are within the paper and its Supporting Information files.

**Funding:** This work was supported by the University of Gondar internal competitive grant awarded in 2020 (R.No. R/T/T/C/E/C/D/03/2013). Debasu Damtie was supported by Sustainable One Health Research Training Capacity (OHEART): Molecular epidemiology of zoonotic foodborne and waterborne pathogens in Eastern Africa, funded by the NIH Fogarty International Center (D43TW008650), through the Global One Health initiative (GOHi). The funders had no role in study design, data collection and analysis, decision to publish, or preparation of the manuscript.

**Competing interests:** The authors have declared that no competing interests exist.

Furthermore, the kit could be used in the evaluation and monitoring of rotavirus vaccine effectiveness in the aforementioned settings.

## Introduction

Rotavirus is the leading cause of severe gastroenteritis among infants and under-five children worldwide. Globally, rotavirus infection was the leading cause of diarrheal deaths, accounting for 19.11% of deaths from diarrhea in 2019 [1]. It is estimated that more than 80% of all rotavirus-related deaths occur in resource-limited countries in South Asia and sub-Saharan Africa [2]. In Ethiopia, rotavirus results in over 28,000 deaths of among under-five children each year accounting for 6% of all rotavirus (RV) deaths globally [3, 4]. Early diagnosis of rotavirus infection minimizes its spread, prevents the unnecessary use of antibiotics, and helps to determine the appropriate treatment [5]. The etiological diagnosis of rotavirus diarrhea requires laboratory confirmation. However, in settings where laboratory tests for rotavirus are not easily accessible, healthcare workers often depend on the analysis of clinical signs and more time-consuming laboratory tests to rule out bacterial causes of diarrhea [6]. The mainstay of management of rotavirus-associated diarrhea is rapid intravenous or oral rehydration therapy [7, 8]. Empirical use of antibiotics for diarrhea treatment without proper identification of the etiological agent is common in resource limited healthcare settings [9].

Clinically, rotavirus gastroenteritis is presented with profuse diarrhea, mild fever, and vomiting, leading to mild-to-severe dehydration [10, 11]. The clinical manifestations of rotavirus diarrhea alone are not sufficiently distinctive to establish diagnosis because of the nonspecific nature of the clinical presentation. Considering the severity of rotavirus diarrhea in children, it is desirable to evaluate and introduce rapid, easy, and cost-effective methods for the detection of rotavirus infection in children. Various laboratory techniques such as electron microscopy, antigen detection immunoassays, molecular assays, and virus isolation can be used to diagnose rotavirus acute gastroenteritis. Electron microscopy (EM) [12] has been used for the diagnosis of rotavirus gastroenteritis since its discovery. However, EM is labor intensive, expensive, requires extensive training and has low sensitivity [13]. Enzyme linked immunosorbent assay (ELISA) has been widely used and its performance is considered satisfactory compared to electron microscopic results but requires specific reagents and equipment. Later, molecular techniques such as reverse transcription—polymerase chain reaction (RT-PCR) replaced other diagnostic tests with the advantage of higher sensitivity and specificity [14]. Although RT-PCR is highly sensitive and specific result, it is labor intensive, costly, requires expertise and hence is not suited for routine laboratory diagnosis particularly in resource-limited settings like Ethiopia. Hence, the use of RT-PCR is limited to research laboratories to detect the viral genome [15, 16].

Currently, several simple and rapid immunochromatographic test kits are commercially available for detection of rotaviruses in stool samples [17, 18]. Most of these tests have high sensitivity and specificity ranging from 90% to 95% [19]. Despite the availability of different platforms for the diagnosis of acute rotavirus gastroenteritis, no single diagnostic tool has been evaluated and approved for use in Ethiopian health care settings. Hence, the current study was undertaken to evaluate the diagnostic performance of the EpiTuub® Fecal Rotavirus Antigen Rapid Test Kit for rotavirus A infection in Ethiopia using one-step RT-PCR as a gold standard.

## Materials and methods

### Study design and settings

A multi-center cross-sectional study design was employed involving three referral hospitals (University of Gondar, l, Felege Hiwot and Debre Markos Comprehensive Specialized Referral Hospitals) in Amhara National Regional State, Ethiopia. The minimum sample size required for the study was determined using single population proportion formula [20].

$$ n = \frac{(Z\alpha/2)^2 P(1-P)}{d^2} $$

Where n = sample size required; Zα/2 = is the critical value of the normal distribution at α/2 for a 95% confidence level, α is 0.05 and the critical value is 1.96, P = proportion of rotavirus among under five children with acute gastroenteritis (25%) [21], and d = margin of error (4%). Considering 15% non-response rate, a total of 537 under-five children with diarrhea were enrolled into the study and stool samples were collected from each study participant between February 1, 2021 and December 31, 2022. Socio-demographic and clinical data of the study participants were collected by trained nurses using semi-structured questionnaires.

### Inclusion and exclusion criteria

All under-five children with acute gastroenteritis visiting the pediatrics departments of the three referral hospitals during the study period were included in the study. Under-five children with acute gastroenteritis who were not able to produce stool during sample collection or/and those who have received rotavirus vaccine (Rotarix) within two weeks of sample collection were excluded from the study.

### Sample collection, transport, and storage

For solid or formed stool 2g of stool was collected while in diarrheic or watery stool, 2ml of stool sample was collected from each under-five diarrheic children in a stool cup and immediately stored in a 2ml cryovial at -20˚C onsite until transported by a cold box to Immunology and Molecular Biology Laboratory at the School of Biomedical and Laboratory Sciences, University of Gondar for laboratory analyses. The samples were then stored at -80˚C at the Immunology and Molecular Biology Laboratory until the laboratory assays were done.

### Rapid fecal rotavirus antigen test

All samples stored at -80˚C were tested for the presence of Rotavirus A antigen using Epi-Tuub® Fecal Rotavirus Antigen Rapid Test Kit (KTR-917, Epitope Diagnostics, San Diego USA) that targets viral protein 6 (VP6) antigen of rotavirus A according to the manufacturer's instruction [22]. In brief, the stool samples were thawed at room temperature and briefly vortexed to uniformly suspend the sample before testing. A small amount of the sample was added to the sampling tube using a sampling lid (the amount that stuck to the sampling lid when inserted 2 times in the sample container) or pipet (3 drops) for solid and liquid samples, respectively. The tube content was then mixed to obtain liquid suspension of the stool sample. After that, the tube was left for ~1 minute to let the debris sediment, the bottom of the test strip was placed into the suspension in a vertical position, and results were read and interpreted within 10 minutes. The presence of red line in both test and control areas is interpreted as positive; the presence of red line only in the control area is interpreted as negative; and the absence of red line in the control area regardless of the test area is considered as invalid test result (Fig 1).

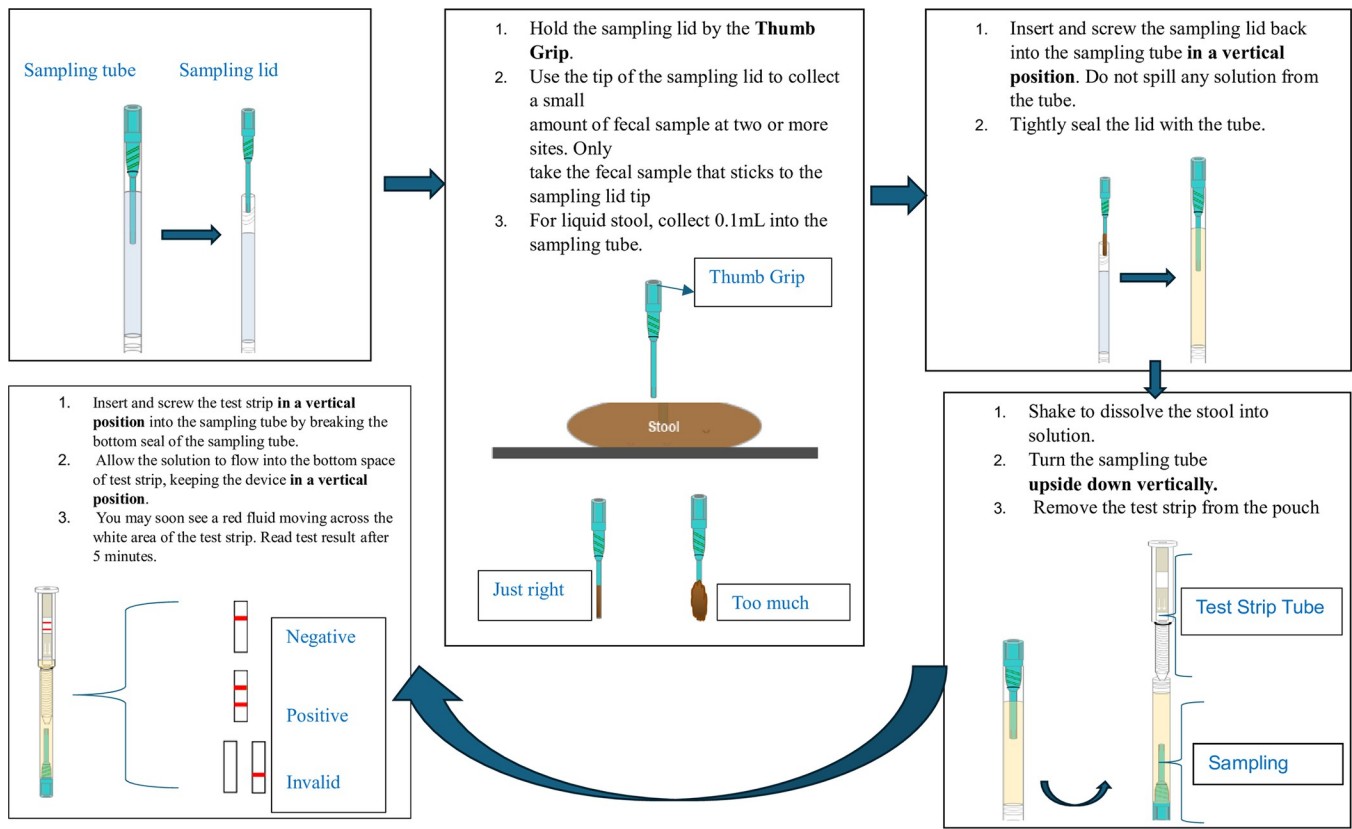

**Fig 1. Rotavirus antigen testing procedure using EpiTuub® Fecal Rotavirus Antigen Rapid Test Kit [23].**

## One-step RT-PCR test for rotavirus detection

Total RNA was extracted using QIAamp Mini spin viral RNA extraction kit (Qiagen, Hilden, Germany, Cat# 52906). RT-PCR was conducted using iTaq Universal SYBR Green One-Step Kit (Bio-Rad, Hercules, California, United States, Cat # 1725151) and HRV NSP3F-5'-ACCATCTACACATGACCCTC-3' and HRV NSP3R-5'-GGTCACATAACGCCCC-3' primers targeting non-structural protein 3 (NSP3) gene of rotavirus A. A total of 18 μl of one-step RT-PCR master mix containing nuclease free water (4.75 μl), (2x) iTaq Universal SYBR Green reaction mix (10 μl), iScriptRT enzyme (0.25 μl), HRV NSP3F primer 10 μM (1.5 μl), and HRV NSP3R primer 10 μM (1.5 μl) was added to each well in a PCR plate. Then, 2 μl of RNA extracted from stool samples was added to each sample-well bringing the total volume to 20 μl. Additionally, 2 μl of positive control, negative extraction control and nuclease free water was added to the respective control wells, containing 18 μl of RT-PCR master mix. RT-PCR settings included a hold step at 50°C for 30 minutes for reverse transcription followed by another hold at 95°C for 10 minutes for initial denaturation and 40 cycles of denaturation at 94°C, annealing at 56°C and extension at 72°C for 30 seconds each. Melting curve analysis was conducted at the end of the RT-PCR program (denaturation at 95°C, annealing 56°C and denaturation at 95°C for 15 seconds each) to ensure the absence of non-specific amplification products including primer-dimers. A specific amplification with cycle threshold (CT) value of less than 40 was considered positive for rotavirus A infection [24, 25].

## Data entry and analysis

The sociodemographic, clinical and laboratory data were entered into SPSS version 29 statistical software for data analysis. The diagnostic performance of the EpiTuub® Fecal Rotavirus Antigen Rapid Test Kit was evaluated against the one-step RT-PCR. The sensitivity, specificity, positive and negative predictive values, and overall test accuracy of the rapid test kit with a 95% confidence interval was calculated. Moreover, the kappa agreement of the two methods was determined. According to Cohen's classification, the kappa value ≤ 0 indicates no agreement, 0.01–0.20 as none to slight agreement, 0.21–0.40 as fair agreement, 0.41–0.60 as moderate agreement, 0.61–0.80 as substantial agreement, and 0.81–1.00 as almost perfect agreement [26]. The receiver operators' curve (ROC) analysis was done and area under the ROC curve (AUC) was determined to assess the overall performance of the test in terms of discriminating rotavirus-infected and noninfected children accurately. The AUC of 0.5 suggests as no discrimination, 0.7–0.8 is considered as acceptable, 0.8–0.9 is considered as excellent and above 0.9 is considered as outstanding [27].

## Ethical considerations

The protocol was approved by the University of Gondar Institutional Review Board (IRB) (R. No. V/P/RCS/05/538/2020). Moreover, hospitals involved in this study were communicated through written letters obtained from the University of Gondar research and publication office (RPO). Written informed consent was obtained from the parents/guardians of children before enrolment to the study. Each study subject/parent/guardian was informed about the objectives of the study and their rights to withdraw from the study at any time. Data taken from study subjects were numerically coded, and the test results were used only for the study purpose and kept securely stored throughout the study period and thereafter. Authors had no access to information that could identify individual participants during or after data collection.

## Results

### Sociodemographic and clinical characteristics

A total of 537 children with diarrhea were enrolled in the study. The mean age of the study participants was 25.5±15.2 months. The majority (97.6%) of the study participants were immunized against rotavirus (Table 1).

### Overall test positivity rate and diagnostic performance characteristics

Among the study participants, 79 (14.71%) and 94 (17.5%) were positive for rotavirus with the rapid antigen test and One-step RT-PCR, respectively. The rapid fecal rotavirus antigen test kit is 75.53% (95% CI = 71.9–79.2) sensitive, 98.19% (95% CI = 97.1–99.3) specific with a positive and negative predictive value of 89.9% (95% CI = 87.3–92.4) and 95% (95% CI = 93.1–96.3), respectively (Table 2).

Five percent (23/458) and 10.12% (8/79) of the results generated by the kit were false-negative and false-positive, respectively (Table 3). The kapa agreement of the rapid fecal rotavirus antigen test kit with one-step RT-PCR was 78.7% (p = 0.0001).

### Receiver operating characteristic curve (ROC) curve analysis

The overall accuracy of the rapid rotavirus antigen test kit is excellent with the area under the ROC curve of 86.9% (95% CI = 81.6, 92.1%) (p = 0.0001) (Fig 2).

**Table 1. Sociodemographic and clinical characteristics of study participants, Ethiopia, 2022.**

| Variables | Number of samples (%) | Rotavirus RT-PCR Positive (%) | Rapid antigen test positive (%) |
|---|---|---|---|
| | N = 537 | N = 94 | N = 79 |
| **Age in months** | (mean ± SD) 25.5±15.2 | | |
| **Sex** | | | |
| Male | 308 (57.4) | 50 (53.2) | 38 (48.1) |
| Female | 229 (42.6) | 44 (46.8) | 41 (51.9) |
| **Residence** | | | |
| Urban | 494 (92) | 89 (91.8) | 74 (93.7) |
| Rural | 43 (8) | 5 (8.2) | 5 (6.3) |
| **Immunization** | | | |
| Yes | 524 (97.6) | 93 (98.9) | 79 (100) |
| No | 13 (2.4) | 1 (1.1) | - |
| **Admission status** | | | |
| Yes | 41 (7.6) | 10 (10.6) | 10 (12.7) |
| No | 496 (92.4) | 84 (89.4) | 69 (87.3) |
| **Dehydration status** | | | |
| No dehydration | 496 (92.4) | 84 (89.4) | 69 (87.3) |
| Some dehydration | 40 (7.4) | 10 (10.6) | 10 (12.7) |
| Severe dehydration | 1 (0.2) | - | - |
| **Intestinal Parasitosis** | | | |
| Positive | 93 (17.3) | 12 (12.8) | 10 (12.7) |
| Negative | 444 (82.7) | 82 (87.2) | 69 (87.3) |

SD = Standard deviation; RT-PCR = Reverse transcriptase polymerase chain reaction; N = Number

## Discussion

Early diagnosis and appropriate management of rotavirus infection in health care facilities and during outbreaks in communities are especially important in sub-Saharan African countries where rotavirus is still the leading cause of acute gastroenteritis and high mortality among under-five children [28]. Additionally, early diagnosis of rotavirus infection with an effective point of care test (POCT) may reduce improper antibiotic use and prevent further development of antibiotic-resistant organisms [5, 29].

In this study, we evaluated the diagnostic performance of the EpiTuub® Fecal Rotavirus Antigen Rapid Test Kit against one-step RT-PCR. In our study, the rapid fecal antigen test kit was found to have a sensitivity of 75.5% and a specificity of 98.2% with 23/458 (5%) false-negative and 8/79(10.12%) false-positive results. These are considered as acceptable performance characteristics in terms of identifying children with the disease as positive and children

**Table 2. The diagnostic performance of EpiTuub® Fecal Rotavirus Antigen Rapid Test Kit, Ethiopia, 2022.**

| Test performance parameter | Performance (%) | 95% CI |
|---|---|---|
| Sensitivity | 75.5 | 71.9–79.2 |
| Specificity | 98.2 | 97.1–99.3 |
| Positive predictive value | 89.9 | 87.3–92.4 |
| Negative predictive value | 95.0 | 93.1–96.8 |
| Prevalence | 17.5 | 14.3–20.7 |

CI = confidence interval

**Table 3. Cross-tabulation for EpiTuub® Fecal Rotavirus Antigen Rapid Test Kit and one step RT-PCR test results, Ethiopia, 2022.**

| Rapid fecal rotavirus A Ag-Test | One-step RT-PCR | | |
|---|---|---|---|
| | Positive | Negative | Total |
| **Positive** | 71 | 8 | 79 |
| **Negative** | 23 | 435 | 458 |
| **Total** | 94 | 443 | 537 |

RT-PCR = Reverse transcriptase polymerase chain reaction, Ag = Antigen

without the disease as negative, respectively [30]. As there are no previous studies evaluating the performance of the rapid test kit used in the current study, we compared our findings with the test performance data of other rapid rotavirus test kits. A study done in Lebanon reported wide variability in sensitivity and specificity among different rapid rotavirus diagnostic kits. According to the Lebanon study, the best performing kit was SD Bioline® with a sensitivity of 95.08% and a specificity of 86.62% and the poorly performing kit was Acon® with a sensitivity of 52.3% and specificity of 10.9% [18].

On the other hand, our study findings are consistent with a study conducted to compare the performance of 3 commercially available enzyme immunoassay kits: Premier™

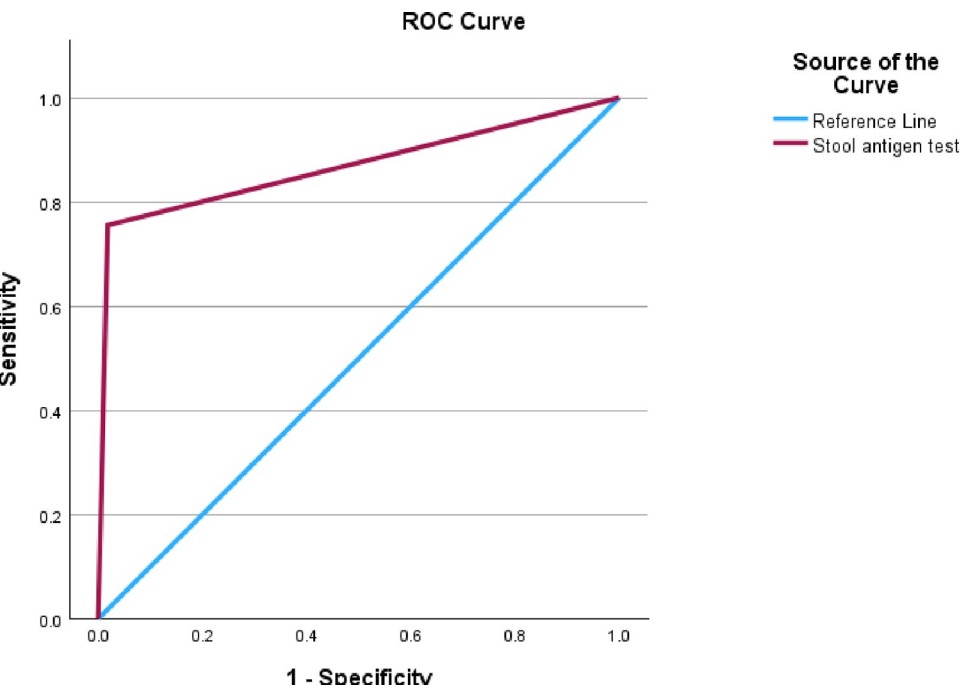

Area under ROC curve =0.869 (95% CI=0.816, 0.921%) (p=0.0001)

**Fig 2. Receiver operating characteristic (ROC) curve for EpiTuub® Fecal Rotavirus Antigen Rapid Test Kit.** The x-axis represents false positive rate (1 –specificity) and y-axis represents true positive rate (sensitivity). The diagonal blue line represents random classification or reference line. Thus, ROC curve is a plot of a test's sensitivity vs. (1-specificity) as well. The closer the point on the ROC curve to the ideal coordinate, the more accurate the test is. The closer the points on the ROC curve to the diagonal, the less accurate the test is. The AUC measures the entire two-dimensional area underneath the entire ROC curve from (0,0) to (1,1). AUC = 0.5 represents a random classifier while AUC = 1.0 represents a perfect classifier.

Rotaclone®, ProSpecT™, and RIDASCREEN® for rotavirus diagnostics. The sensitivity of the kits ranged from 75% to 82.1% while the specificity was 100% for all the evaluated kits [31]. Another study reported that VIKIA® Rota-Adeno rapid test kit had high specificity (91.6%) but low sensitivity (44.8%) of rotavirus detection in clinical settings [32]. A study conducted in refugee camps on the border between Kenya and Somalia reported comparable findings of sensitivity (83.1%) and specificity (99.3%) for ImmunoCard STAT!® Rapid Diagnostic Test kit [33].

In our study, the kit has also shown 89.87% positive predictive value (PPV) and 94.98% negative predictive value (NPV) which makes this kit a great option to be used as a point of care test kit to quickly diagnose rotavirus infections. A similar finding was reported form the Lebanese study for SD Bioline® with PPV of 95.08% and NPV of 86.62%. However, this same Lebanese study reported 40.96% PPV and 16.21% NPV for Acon® RDT [18]. High positive (98.3%) and negative (92.1%) predictive values were reported from a study conducted at a refugee camp for ImmunoCard STAT!® Rapid Diagnostic Test kit [33]. The variability in the diagnostic performance of the rapid test kits may result from the inherent variation associated with the use of different target antigens by manufacturing companies of the different test kits. Moreover, some studies use one-step RT-PCR as a gold standard as we did while others such as the study in Kenya and Somalia (evaluating ImmunoCard STAT! ® Rapid Diagnostic Test kit) have used ELISA as a reference test unlike our study.

The kapa agreement of the rapid fecal rotavirus antigen test kit with one step RT-PCR was 78.7% representing a substantial agreement between the two methods [26, 34]. The area under the ROC curve (AUR) of the evaluated kit was 86.9% (95% CI = 81.6, 92.1%), which is interpreted as excellent overall accuracy of the test kit [27].

Results were generated and interpreted within 10 minutes, which would provide a rapid point of care test result highly desired for children with acute gastroenteritis. The kit does not require proprietary equipment to perform the test which makes it easily applicable in the resource limited settings like in Ethiopia. The test kit requires a cold box for transportation and a refrigerator for storage at a temperature of 2–8˚C. This is achievable in the primary health care settings of Ethiopia where refrigerators are dedicated to store vaccines and other biologicals that require refrigeration for their storage. Thus, the EpiTuub® Fecal Rotavirus Antigen Rapid Test Kit is a promising approach to overcoming practical challenges such as the need to batch samples, the long turnaround time, and the high costs associated with rotavirus testing using one step RT-PCR methods. This kit fulfils most of the WHO criteria for POCT summarized in its acronym ASSURED (Affordable, Sensitive, Specific, User-Friendly, Rapid, Equipment-Free, Delivered) [35, 36]. Hence, it can be used as point-of-care rotavirus diagnostic tool with good utility in Ethiopian health care settings and similar resource-limited or remote settings.

## Limitations of the study

The study used only one kit for evaluation. However, it would have been good to include multiple rapid rotavirus test kits for evaluation so that the best performing kit can be considered for use in resource limited settings.

## Conclusions and recommendation

The EpiTuub® Fecal Rotavirus Antigen Rapid Test Kit (KTR-917, Epitope Diagnostics, San Diego USA) is a sensitive, specific, user-friendly, rapid, and equipment-free option to be used as a POCT in Ethiopian healthcare settings where resource is limited to do one-step RT-PCR. Furthermore, the kit could be used in the evaluation and monitoring of rotavirus vaccine

effectiveness in such settings. Therefore, we recommend the Ministry of Health of Ethiopia and Ethiopian Public Health Institute to consider the kit for the diagnosis of rotavirus infection among under-five children in Ethiopian healthcare system. We would also like to recommend further diagnostic performance evaluation study comparing several commercially available rapid rotavirus test kits to broaden options in terms of cost and availability.

## Supporting information

**S1 Checklist. STROBE statement—checklist of items that should be included in reports of observational studies.**
(DOCX)

## Acknowledgments

We would like to acknowledge the University of Gondar, the Ohio State University and NIH-Fogarty international center for their support. We are also grateful to the data and sample collectors, and study participants.

## Author Contributions

**Conceptualization:** Debasu Damtie, Anastasia N. Vlasova, Belay Tessema.

**Data curation:** Debasu Damtie, Yetemwork Aleka, Anastasia N. Vlasova, Belay Tessema.

**Formal analysis:** Debasu Damtie, Yetemwork Aleka, Zewdu Siyoum Tarekegn, Anastasia N. Vlasova, Belay Tessema.

**Funding acquisition:** Yitayih Wondimeneh.

**Investigation:** Debasu Damtie, Yetemwork Aleka, Zewdu Siyoum Tarekegn.

**Methodology:** Debasu Damtie, Anastasia N. Vlasova, Belay Tessema.

**Project administration:** Debasu Damtie.

**Resources:** Ulrich Sack, Anastasia N. Vlasova.

**Supervision:** Aschalew Gelaw, Yitayih Wondimeneh, Ulrich Sack, Anastasia N. Vlasova, Belay Tessema.

**Validation:** Debasu Damtie.

**Writing – original draft:** Debasu Damtie.

**Writing – review & editing:** Debasu Damtie, Aschalew Gelaw, Yitayih Wondimeneh, Yetemwork Aleka, Zewdu Siyoum Tarekegn, Ulrich Sack, Anastasia N. Vlasova, Belay Tessema.

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
