## [Decision Letter · Decision Letter 0]

10 Oct 2023

PONE-D-23-19314Evaluation of the Diagnostic Performance of EpiTuub® Fecal Rotavirus Antigen Rapid Test Kit in Amhara National Regional State, Ethiopia: A multi-center cross-sectional studyPLOS ONE

Dear Dr. Damtie,

Thank you for submitting your manuscript to PLOS ONE. After careful consideration, we feel that it has merit but does not fully meet PLOS ONE’s publication criteria as it currently stands. Therefore, we invite you to submit a revised version of the manuscript that addresses the points raised during the review process.

We look forward to receiving your revised manuscript.

Kind regards,

Enoch Aninagyei, PhD

Academic Editor

PLOS ONE

A clean copy of the edited manuscript (uploaded as the new *manuscript* file)”.

Reviewers' comments:

Reviewer's Responses to Questions

**Comments to the Author**

1. Is the manuscript technically sound, and do the data support the conclusions?

Reviewer #1: Yes

Reviewer #2: Partly

2. Has the statistical analysis been performed appropriately and rigorously? 

Reviewer #1: Yes

Reviewer #2: I Don't Know

3. Have the authors made all data underlying the findings in their manuscript fully available?

Reviewer #1: Yes

Reviewer #2: Yes

4. Is the manuscript presented in an intelligible fashion and written in standard English?

Reviewer #1: Yes

Reviewer #2: Yes

5. Review Comments to the Author

Reviewer #1: The authors explore “Evaluation of the Diagnostic Performance of EpiTuub® Fecal Rotavirus Antigen Rapid Test Kit in Amhara National Regional State, Ethiopia: A multi-center cross-sectional study”. However, the study was needed to some modifications as following to be ready for publications. Before it could be accepted, please address the following comments in your consideration:

1-The comparison between the Rotavirus A antigen using EpiTuub® 107 Fecal Rotavirus Antigen Rapid Test Kit (KTR-917, Epitope Diagnostics, Sandiago USA) and One-step RT-PCR is good, as the sensitivity rate reaches 98%, But in order to get a real comparison, this study must be compared with other available kits. In addition to, Some language mistakes and grammar mistakes I advise the revising for English Language by a native English speaker.

2-Why the authors did not mentioned the catalogue number of QIAamp Mini spin viral RNA extraction kit and One step RT-PCR kit?

3-What the type of virus which used as positive control in the different method?

The manuscript should be revise d based on the comments raised above

With my best wishes

Fatma Abdallah

Reviewer #2: Concerning Question 1

a. Causes of diarrheal diseases can be polymicrobial and the diagnosis of rotavirus does not exclude other viral or bacterial causes. The focus of the study was on rotavirus diagnosis using EpiTuub® Fecal Rotavirus Antigen Rapid Test Kit . I suggest the authors play down on statements that implies that rotavirus diagnosis excludes bacterial causes of diarrhea.

b. Table 1 mentioned the immunization status of the participants with regards to the Rotavirus RT-PCR results. The status of the participants as regards the rotavirus antigen test kit findings was not mentioned in the results or subsequent discussion. Hence, there is no basis from this study to conclude that the kit could be used in the evaluation and monitoring of rotavirus vaccine effectiveness

Concerning Question 2

The authors should elaborate on how their sample size of 537 was derived

Ethical consideration: What was the nature of the written communication from the University of Gondar research and publication office to the three referral hospitals

6. PLOS authors have the option to publish the peer review history of their article (what does this mean?). If published, this will include your full peer review and any attached files.

Reviewer #1: No

Reviewer #2: No

---

## [Author Response · Author response to Decision Letter 0]

26 Oct 2023

Response to the editor and the reviewers

First of all, we would like to thank the editor and the reviewers of our manuscript for the invaluable and highly relevant comments which helped us to improve our manuscript. Below is our point-by-point response to the editor’s and the reviewer’s comments:

Response to the editor: 

• RESPONSE: Thank you dear editor. We have considered the comment and edited the file naming styles other journal styles as per the template provided in the link. 

• RESPONSE: Thank you dear editor. We have made a language edition by Dr. Anastasia N. Vlasova (Associate Professor at The Ohio State University), one of the most senior members of the authors as per your recommendation. 

Response to the reviewers:

Reviewer #1

1. The comparison between the Rotavirus A antigen using EpiTuub® 107 Fecal Rotavirus Antigen Rapid Test Kit (KTR-917, Epitope Diagnostics, Sandiago USA) and One-step RT-PCR is good, as the sensitivity rate reaches 98%, But in order to get a real comparison, this study must be compared with other available kits. In addition to, Some language mistakes and grammar mistakes I advise the revising for English Language by a native English speaker.

• RESPONSE: Thank you dear reviewer, as you mentioned it would have been better if we had evaluated as many kits as possible to identify the best performing kit among many. However, the scope of our study was to evaluate the EpiTuub® 107 Fecal Rotavirus Antigen Rapid Test Kit (KTR-917, Epitope Diagnostics, Sandiago USA) against one-step RT-PCR due to limited resources we have. Hence, we have indicated it as a limitation of this study and recommended further study involving as many kits as possible. As for the language edition, we have made language edition by one of the most senior members of the research team in the revised version of the MS. 

2. Why the authors did not mentioned the catalogue number of QIAamp Mini spin viral RNA extraction kit and One step RT-PCR kit?

• RESPONSE: Dear reviewer, thank you for the comment. The Catalogue number of the QIAmp mini spin viral RNA extraction kit was (Qiagen, Hilden, Germany, Cat # 61904) and One step RT-PCR kit ((Bio-Rad, Hercules, California, United States‎, Cat # 1725151). We have indicated the catalogue number of the kits in the methods section of the revised version of the manuscript. 

3. What the type of virus which used as positive control in the different method?

• RESPONSE: We have used rotavirus RNA extracted from known rotavirus A positive stool sample which was deposited at -80 oc in Immunology and Molecular Biology Laboratory of the University of Gondar as a positive control for the one-step RT-PCR. 

Reviewer #2

a. Causes of diarrheal diseases can be polymicrobial and the diagnosis of rotavirus does not exclude other viral or bacterial causes. The focus of the study was on rotavirus diagnosis using EpiTuub® Fecal Rotavirus Antigen Rapid Test Kit. I suggest the authors play down on statements that implies that rotavirus diagnosis excludes bacterial causes of diarrhea.

• RESPONSE: Dear reviewer, thank you for the comment. It is true that the cause of diarrhea is polymicrobial. Hence, investigation should involve all possible causes of diarrhea including viral causes. The current practice in Ethiopia is however focused only on investigation of parasitic and bacterial causes of diarrhea ignoring the viral causes of diarrhea. Of the viral causes, rotavirus is the leading cause of diarrhea and diagnosing rotavirus in addition to routine parasitic and bacterial investigations could help reduce empirical treatment of diarrhea in the study settings. Which in turn reduces the risk of antimicrobial resistance. Otherwise, we did not conclude that rotavirus diagnosis by itself rule out bacterial causes of diarrhea.

b. Table 1 mentioned the immunization status of the participants with regards to the Rotavirus RT-PCR results. The status of the participants as regards the rotavirus antigen test kit findings was not mentioned in the results or subsequent discussion. Hence, there is no basis from this study to conclude that the kit could be used in the evaluation and monitoring of rotavirus vaccine effectiveness

• RESPONSE፡ Thank you dear reviewer for the comment. We have considered your comment and included the results for the rapid fecal rotavirus antigen test kit on Table 1. Of course, our focus was to evaluate the performance of the rapid rotavirus antigen test kit using one step RT-PCR as a gold standard. Accordingly, we have presented the test performance characteristics of the rapid test kit in the results section and discussed the findings. The test performance characteristics of the rapid antigen test kit in comparison to the gold standard was found to be acceptable. Hence, the test kit can be used to diagnose rotavirus infection among children. The WHO recommends use of a “case test-negative control” epidemiological study design for vaccine effectiveness studies in resource limited settings. Hence, our conclusion on the use of the kit for vaccine effectiveness studies is based on the aforementioned WHO’s recommendation in which the kit can be used to screen rotavirus positive cases and rotavirus negative controls (test negative controls) and trace back their immunization status retrospectively to determine rotavirus vaccine effectiveness in resource limited settings. 

c. Concerning Question 2, The authors should elaborate on how their sample size of 537 was derived

• RESPONSE: Considering the single population proportion formula, 

 n = (Zα/2)2 P(1-P) 

 d2 

Where n = sample size required; Zα/2 = standard normal variate for level of confidence (95%, 1.96), P = proportion of rotavirus among under five children with acute gastroenteritis (25%) (according to a previous study from reference 34); and d = margin of error (4%). We have also considered 10% non-response rate to get the final minimum sample size required for the study. Based on this formula, the minimum sample size calculated was 495. However, the hospitals involved in the study extended a little bit and collected 537 samples for our study. Hence, we considered all the 537 study participants to maximize the precision of the estimate. 

d. Ethical consideration: What was the nature of the written communication from the University of Gondar research and publication office to the three referral hospitals

• RESPONSE: The letter written from the University of Gondar research and publication office explains the objectives and the relevance of the study and requests the hospitals for permission and cooperation in the study. We have indicated the nature of the letter in the revised version of the manuscript.

---

## [Editor Report · Decision Letter 1]

7 Nov 2023

PONE-D-23-19314R1Evaluation of the diagnostic performance of EpiTuub® Fecal Rotavirus Antigen Rapid Test Kit in Amhara National Regional State, Ethiopia: A multi-center cross-sectional study

PLOS ONE

Dear Dr. Damtie,

Thank you for submitting your manuscript to PLOS ONE. After careful consideration, we feel that it has merit but does not fully meet PLOS ONE’s publication criteria as it currently stands. Therefore, we invite you to submit a revised version of the manuscript that addresses the points raised during the review process.

ACADEMIC EDITOR:

Line 25: antibiotics are not used to treat viral infections

Lines 87-89: Revise it to read University of Gondar, Felege Hiwot and Debre Markos Comprehensive Specialized Referral Hospitals

Lines 89-90: indicate in detail how the sample size was arrived at

Line 94-100: Any inclusion and exclusion criteria?

Line 95: indicate that for solid or formed stool 2g of stool was collected while in diarrheic or watery stool, 2ml of stool sample was collected

Line 96: was the cryovial not overfilled? How did you placed a 2ml watery stool in a 2ml vial?

Line 98: change ‘analysis’ to ‘analyses’

Line 99: change ‘assay’ to ‘assays’

Line 105/109: how did you homogenized the samples? Or you mean the samples were uniformly suspended?

Lines 105-111: The EpiTuub® Fecal Rotavirus Antigen Rapid Test Kit looks novel in your settings. Therefore, it is important to use schematic diagrams to describe how the test was performed. Visit this site to see example Strep A Rapid Test Cassette (Control Line in Blue) JusChek Infectious Disease Rapid Test Kuala Lumpur (KL), Malaysia, Selangor Supplier, Suppliers, Supply, Supplies | Setia Scientific Solution

Lines 131-133: A specific amplification with cycle threshold (CT) value of less than 40 was considered positive for rotavirus A infection. Provide reference for this statement.

Line 144: Bring ‘Data entry and analysis’ before ‘Ethical considerations’

Tables 1-3: Headings look too long. Summarize

We look forward to receiving your revised manuscript.

Kind regards,

Enoch Aninagyei, PhD

Academic Editor

PLOS ONE
---

## [Author Response · Author response to Decision Letter 1]

15 Nov 2023

Response letter to the editor and reviewers

PONE-D-23-19314R1

Evaluation of the diagnostic performance of EpiTuub® Fecal Rotavirus Antigen Rapid Test Kit in Amhara National Regional State, Ethiopia: A multi-center cross-sectional study

We would like to thank the editor and the reviewers of our manuscript for the invaluable and highly relevant comments which helped us to improve our manuscript. Below is our point-by-point response to the editor’s and the reviewer’s comments:

ACADEMIC EDITOR:

Line 25: antibiotics are not used to treat viral infections

Response: Thank you, dear Editor, we rewrite the sentence as per your comment. 

Lines 87-89: Revise it to read University of Gondar, Felege Hiwot and Debre Markos Comprehensive Specialized Referral Hospitals

Response: thank you dear Editor, we have considered your comment and modified the manuscript accordingly.

Lines 89-90: indicate in detail how the sample size was arrived at

Response: Thank you, dear Editor, we have included how we determined the sample size in the revised manuscript.

Line 94-100: Any inclusion and exclusion criteria?

Response: Thank you, dear Editor, we have included the inclusion and exclusion criteria in the revised manuscript.

Line 95: indicate that for solid or formed stool 2g of stool was collected while in diarrheic or watery stool, 2ml of stool sample was collected

Response: Thank you, dear Editor, we have considered your suggestions in the revised manuscript. 

Line 96: was the cryovial not overfilled? How did you placed a 2ml watery stool in a 2ml vial?

Response: We have collected approximately 2ml watery stool in a stool cup and transferred it into a 2 ml cryovial for storage. However, when we transfer the samples for storage, we added slightly lower volume than two ml to avoid overfilling of the cryovials. 

Line 98: change ‘analysis’ to ‘analyses’

Response: We have accepted the change and considered it in the revised manuscript.

Line 99: change ‘assay’ to ‘assays’

Response: We have accepted the change and considered it in the revised manuscript. 

Line 105/109: how did you homogenized the samples? Or you mean the samples were uniformly suspended?

Response: Dear Editor, it is just to mean samples were uniformly suspended. We have modified the statement to avoid any confusion.

Lines 105-111: The EpiTuub® Fecal Rotavirus Antigen Rapid Test Kit looks novel in your settings. Therefore, it is important to use schematic diagrams to describe how the test was performed. Visit this site to see example Strep A Rapid Test Cassette (Control Line in Blue) JusChek Infectious Disease Rapid Test Kuala Lumpur (KL), Malaysia, Selangor Supplier, Suppliers, Supply, Supplies | Setia Scientific Solution

Response: Dear Editor, we have adapted the testing procedure from the kit insert leaflet and included schematic diagram to show the testing procedure of the kit. 

Lines 131-133: A specific amplification with cycle threshold (CT) value of less than 40 was considered positive for rotavirus A infection. Provide reference for this statement.

Response: Thank you, dear Editor, we have cited references. 

Line 144: Bring ‘Data entry and analysis’ before ‘Ethical considerations’

Response: Dear Editor, we have moved data entry and analysis section before ethical considerations section as per your considerations. 

Tables 1-3: Headings look too long. Summarize

Response: We have modified and shortened the headings of the tables

---

## [Editor Report · Decision Letter 2]

17 Nov 2023

Evaluation of the diagnostic performance of EpiTuub®Fecal Rotavirus Antigen Rapid Test Kit in Amhara National Regional State, Ethiopia: A multi-center cross-sectional study

PONE-D-23-19314R2

Dear Dr. Debasu Damtie,

We’re pleased to inform you that your manuscript has been judged scientifically suitable for publication and will be formally accepted for publication once it meets all outstanding technical requirements.

Kind regards,

Enoch Aninagyei, PhD

Academic Editor

PLOS ONE
---

## [Editor Report · Acceptance letter]

22 Nov 2023

PONE-D-23-19314R2 

Evaluation of the diagnostic performance of EpiTuub® Fecal Rotavirus Antigen Rapid Test Kit in Amhara National Regional State, Ethiopia: A multi-center cross-sectional study 

Dear Dr. Damtie:

I'm pleased to inform you that your manuscript has been deemed suitable for publication in PLOS ONE. Congratulations! Your manuscript is now with our production department. 

Kind regards, 

on behalf of

Dr Enoch Aninagyei 

Academic Editor

PLOS ONE